# Comprehensive Map of the Regulated Cell Death Signaling Network: A Powerful Analytical Tool for Studying Diseases

**DOI:** 10.3390/cancers12040990

**Published:** 2020-04-17

**Authors:** Jean-Marie Ravel, L. Cristobal Monraz Gomez, Nicolas Sompairac, Laurence Calzone, Boris Zhivotovsky, Guido Kroemer, Emmanuel Barillot, Andrei Zinovyev, Inna Kuperstein

**Affiliations:** 1Institut Curie, PSL Research University, Mines Paris Tech, Inserm, U900, 75005 Paris, France; jean.marie.ravel@gmail.com (J.-M.R.); luis-cristobal.monraz@curie.fr (L.C.M.G.); Nicolas.Sompairac@curie.fr (N.S.); laurence.calzone@curie.fr (L.C.); emmanuel.barillot@curie.fr (E.B.); andrei.zinovyev@curie.fr (A.Z.); 2Laboratoire de génétique médicale, CHRU-Nancy, F-54000 Nancy, France; 3Inserm, NGERE, Université de Lorraine, F-54000 Nancy, France; 4Centre de Recherches Interdisciplinaires, Université Paris Descartes, 75006 Paris, France; 5Faculty of Medicine, Lomonosov Moscow State University, 119991 Moscow, Russia; Boris.Zhivotovsky@ki.se; 6Division of Toxicology, Institute of Environmental Medicine, Karolinska Institutet, Box 210, 17177 Stockholm, Sweden; 7Centre de Recherche des Cordeliers, Equipe labellisée par la Ligue contre le cancer, Université de Paris, Sorbonne Université, Inserm U1138, Institut Universitaire de France, 75006 Paris, France; kroemer@orange.fr; 8Metabolomics and Cell Biology Platforms, Institut Gustave Roussy, 94805 Villejuif, France; 9Pôle de Biologie, Hôpital Européen Georges Pompidou, AP-HP, 75015 Paris, France; 10Suzhou Institute for Systems Medicine, Chinese Academy of Medical Sciences, Suzhou 215163, China; 11Karolinska Institute, Department of Women’s and Children’s Health, Karolinska University Hospital, 171 77 Stockholm, Sweden

**Keywords:** regulated cell death, survival, signaling network, comprehensive map, biocuration, data visualization, module activity, lung cancer, Alzheimer’s disease, NaviCell

## Abstract

The processes leading to, or avoiding cell death are widely studied, because of their frequent perturbation in various diseases. Cell death occurs in three highly interconnected steps: Initiation, signaling and execution. We used a systems biology approach to gather information about all known modes of regulated cell death (RCD). Based on the experimental data retrieved from literature by manual curation, we graphically depicted the biological processes involved in RCD in the form of a seamless comprehensive signaling network map. The molecular mechanisms of each RCD mode are represented in detail. The RCD network map is divided into 26 functional modules that can be visualized contextually in the whole seamless network, as well as in individual diagrams. The resource is freely available and accessible via several web platforms for map navigation, data integration, and analysis. The RCD network map was employed for interpreting the functional differences in cell death regulation between Alzheimer’s disease and non-small cell lung cancer based on gene expression data that allowed emphasizing the molecular mechanisms underlying the inverse comorbidity between the two pathologies. In addition, the map was used for the analysis of genomic and transcriptomic data from ovarian cancer patients that provided RCD map-based signatures of four distinct tumor subtypes and highlighted the difference in regulations of cell death molecular mechanisms.

## 1. Introduction

Cell death garners special attention not only because it represents the endpoint of the cell’s life cycle, but also because the molecular mechanisms leading to cellular demise are perturbed in many different diseases [1,2]. Two major scenarios are sealing a cell’s final fate. Cells can either die from regulated cell death (RCD) or succumb to accidental cell death [3,4]. Accidental or non-regulated cell death, which is often referred to as necrosis, is a sudden process that combines cell membrane rupture and release of the cellular content into the surrounding environment [5]. In contrast, the RCD involves an active contribution by the cells and may be executed through a panoply of different pathways that include (but are not limited to) apoptosis, necroptosis, autophagy, ferroptosis, parthanatos, and pyroptosis [6,7]. 

One of the most studied modes of RCD is apoptosis, a ‘clean’ cell elimination process that has been thought to be non-inflammatory and non-immunogenic. However, in specific circumstances, cells dying from apoptosis may induce an immune response against dead-cell antigens [8]. Apoptosis is a highly regulated, energy-dependent process that requires ATP supply and, therefore, hinges on a functional bioenergetics metabolism. The severity of environmental stress, damage, genetic, or epigenetic perturbations dictate the choice between survival or death program and also mode of cell death execution, be it apoptosis or another lethal subroutine [9]. A challenge is to integrate all the distinct initiation, signaling and execution pathways into one seamless map. RCD is coordinated by numerous signaling pathways including trophic signals relayed by AKT and mTOR, stress kinase including ERK and JNK, or transcription factors such as NF-κB. RCD is also dependent on mitochondria and glucose metabolism that dictate the energy supply status. In addition, there is a complex post-transcriptional regulation level by non-coding RNAs including miRNAs and post-translational modifications of apoptosis-related proteins. Morphologically, apoptosis is characterized by cell shrinkage, nuclear condensation and fragmentation, cellular detachment from surrounding tissue and surface exposure of “eat-me” signals such as phosphatidylserine that facilitate removal of the dying cells by macrophages. 

Another well-studied form of RCD, autophagy, is activated under various stress situations, in particular in response to a shortage in energy supply in the context of starvation or scarcity of growth factors [10]. Autophagy ensures cell homeostasis by recycling cellular organelles and macromolecules, thereby, mediating cytoprotection and reducing the propensity of cells to undergo RCD. In specific circumstances, however, autophagy can be switched to a process of self-elimination leading to cell death. Necroptosis is a signaling-mediated mode of cell death with morphological features similar to necrosis. Necroptosis is accompanied by “swelling” of cells and organelles and rapid loss of plasma membrane integrity. Ferroptosis is a cell death modality induced by an accumulation of lipid peroxides due to a dysregulation of iron metabolism that facilitates the generation of reactive oxygen species (ROS) and consequent peroxidation of membrane lipids [11]. Parthanatos is a Poly [ADP-ribose] polymerase 1 (PARP1) dependent cell death mode induced mostly by intracellular inputs such as DNA damage. When activated by single-stranded DNA breaks, PARP1 synthesizes poly (ADP-ribose) chains (PAR) that are covalently attached to nuclear proteins while consuming ADP and NAD+. PAR then acts as an activating signal for other DNA-repairing enzymes. However, in the case of excessive DNA damage, DNA repair processes are overstimulated, which leads to the consumption of ATP and NAD depletion, triggering cellular execution. Parthanatos is associated with mitochondrial fission, its major phenotype [12]. Finally, pyroptosis is a caspase-1 dependent cell death modality that can be activated by extracellular signals (such as bacterial lipopolysaccharide, LPS) that bind to receptors (such as NLRC4), inducing receptor dimerization that triggers PYCARD protein recruitment and thus caspase-1 activation.

This massive knowledge of RCD mechanisms is scattered through thousands of scientific publications. This situation hampers not only the holistic understanding of RCD regulation but also the application of bioinformatics and systems biology approaches. Efforts of systematic collection and knowledge formalization about molecular interactions in a structured and computer-readable format have already been applied to numerous pathway databases. Several databases contain partially overlapping collections of signaling diagrams representing different modes of cell death, mitochondria and glucose metabolism in some detail [13,14,15,16,17]. However, the limiting factor of this type of signaling representation is the lack of an integrated view of the pathways linking upstream inputs, such as activators of ‘death receptor’, damaging agents or intracellular stressors to the different regulated cell death modes. 

As an additional approach for data formalization, protein-protein interaction (PPI) networks are used to depict reciprocal effects between molecular entities. However, PPI diagrams do not represent interactions between non-protein molecular entities (such as metabolites) and the depth of process description is generally not sufficient to understand the directionality of the signal flow, regulatory effects or feedback circuitries. 

Cell signaling can also be depicted at the level of biochemical reactions in the form of process diagrams, thus creating a comprehensive map as has been done for RB-E2F signaling [18], mTOR signaling [19], Alzheimer’s pathogenesis [20], influenza A virus replication [21], the Atlas of Cancer Signaling Networks (ACSN) [22], and others [23]. 

Here, we present the construction of a comprehensive illustration of interactions among different RCD modes in the form of a seamless signaling network. The resulting map constitutes the first integrated representation of all known modes of RCD together. The map depicts inputs, the initiating factors that activate different cell death mechanisms, the signaling pathways responsible for the choice of distinct lethal subroutines and the final executing mechanisms. These three elements correspond to the three layers of the map. The RCD map also covers related energy, metabolic, and mitochondrial mechanisms. To facilitate navigation and usage, the map is divided into functional modules. This systems biology resource is freely available, allows easy navigation and curation by the community, facilitated by several web-based platforms such as NaviCell [24], MINERVA [25], and NDEx [26]. Finally, we demonstrate that the RCD map can be used as a tool for comparison of cell death regulation in several human diseases. 

## 2. Results

### 2.1. General Characteristics of Regulated Cell Death Map 

The molecular mechanisms are depicted on the map in the form of a biochemical reaction network using a well-established methodology [27]. The diagrams are built using the Systems Biology Graphical Notation language (SBGN) [28] and drawn using the CellDesigner tool [29] that ensures compatibility of the maps with various tools for network analysis, data integration, and network modelling. 

To construct the map, manual literature curation was carefully performed in several steps, first retrieving a level of well-established ‘consensus’ information from major reviews and then adding details from recent original publications (Figure 1 and Appendix A). 

RCD is a tightly regulated process characterized by numerous cross-talks between molecular mechanisms. In order to represent these details in an organized and user-friendly manner, we handled this complexity in two ways. First, the map has a hierarchical structure and is constructed around three successive layers (initiation, signaling, execution). The map is also split into meta-modules, then into functional modules, and, ultimately, the most detailed level describes biochemical reactions (Figure 2 and Table 1). The second strategy to address complexity relies on the introduction of a tagging system for map layers, modules and each entity and reaction that indicate their involvement in different biological processes. This system allows tracing the participation of entities in different processes and helps to retrieve a backbone structure of the network diagram. 

The components of the maps are annotated in the NaviCell format, including PubMed references, cross-references with other databases, notes by the map manager. In addition, each molecular player possesses module tags (Figure 2 and Appendix A) and each reaction is assigned with a confidence score (Appendix A). The RCD map summarizes information 1008 proteins, 260 genes, 93 miRNAs and 2020 reactions. It is based on 800 original scientific publications. The map is divided into three meta-modules and 26 separate functional modules (Table 1). The principles and procedure for map construction, including the graphical standards, data model, literature curation rules, data input from other databases, the detailed tagging system and confidence scores description are provided in the Materials and Methods section.

### 2.2. Structure of the Regulated Cell Death Map

RCD is a dynamic process, wherein the propagation and duration of signals determine cell fate decision. As the name suggests, RCD is highly regulated, in addition, different players may share the same modulators or the same targets. The RCD process can be split into three major steps, initiation, signaling and execution, which we defined as the layers of the map (Figure 2 and Table 1). The “Initiation” layer depicts input signals and mechanisms that ignite RCD. 

The three major inputs are grouped into the “Ligand–Receptor”, “Metabolism”, and “Stress Response” modules. The “Signaling” layer is the level where the decision between cell death and survival is made along with the selection of the cell death mode. It gathers all previously described cell death modes and their inhibition processes. It is also the step of interplay where primary clues transmit their signal to a secondary messenger depending on the cellular context. Thus, the same primary/initiation cue can trigger different types of outcome depending on the absence or presence of other entities. The “Execution” layer represents the final step of RCD in which the process trespasses the point-of-no-return and vital components of the cell are irreversibly degraded. Mitochondrial outer membrane permeabilization (MOMP), mitochondrial permeability transition (MPT) and caspases are the major players of this step. Each module in the “signaling” layer triggers a specific combination of modules in the “execution” layer. For example, initiation of parthanatos triggers large scale DNA fragmentation, whereas ignition of apoptosis preferentially leads to caspases activation and to MOMP. 

In addition, a modular structure of RCD map has been introduced, allowing to explore each molecular mechanism separately, represented as an independent module map. The RCD map is divided into functional meta-modules that are further subdivided into smaller functional modules, together creating 26 functional modules. For example, the meta-module dealing with “death receptor pathways” contains information on the response to TRAIL, the ligation of Fas receptors, the response to TNF, and the signaling of dependence receptors (Table 1). The detailed description of layers, functional meta-modules and modules are found in the Appendix A.

### 2.3. Contents of the RCD Map

The RCD map gathers information on all the major regulated death pathway, facilitating illustration of their cross-talks. 

Apoptosis is classically divided into “extrinsic apoptosis”, depicted at the right of our drawing and “intrinsic apoptosis”, at the left [7]. Extrinsic apoptosis is triggered by death receptors, in which ligand binding triggers the signal, and dependence receptors, in which ligand withdrawal activates the signal [30]. This initiation signal leads to rapid activation of caspases (especially CASP3 and CASP7) and eventually apoptotic death [31]. Intrinsic apoptosis preferentially leads to mitochondrial outer membrane permeabilization (MOMP), often as a result of the oligomerization of BAX/BAK [32] and their insertion into the mitochondrial membrane. Apoptosome-dependent CASP9 activation in the cytoplasm occurs as a result of the mitochondrial release of CYCS (previously known as cytochrome c) and SMAC (also known as DIABLO). CASP9 activation leads to subsequent proteolytic CASP3 and CASP7 activation and apoptosis (Appendix A). 

The Necroptosis outbreak is guided by RIPK1 and RIPK3 proteins [33]. When CASP8 is inhibited, especially by c-FLIP (encoded by CFLAR), RIPK3 is activated by RIPK1. RIPK3 and MLKL recruitment into the necrosome complex triggers the features of necroptosis (a rise in cytosolic Ca^2+^, ROS production, PARP1 over-activation, and lysosome membrane permeabilization) and eventually necroptotic death (Appendix A). 

Autophagy and lysosome-dependent cell death is influenced by the cell energy status and coupled to the formation of autophagosomes and autophagolysosomes [34]. ROS, amino acid starvation, and DNA damage result in a decrease of cytosolic 5′ adenosine triphosphate (ATP) and a consequent increase in 5′ adenosine monophosphate (AMP), thus triggering activation of the AMP-dependent kinase (AMPK). This triggers (1) inhibition of pro-survival complex mTORC1; (2) activation of BECN1 (Beclin 1) and results in autophagosome formation (Appendix A). 

The central feature of parthanatos is the over-activation of PARP1 following alkylating DNA damage or exposure to ROS [12,35]. Excessive activation of PARP1 leads to PAR accumulation, finally triggering AIMF1 release from mitochondria and AIFM1-mediated DNA fragmentation. PARP1 is also responsible for NAD+ and ATP depletion causing a fatal collapse in bioenergetics and redox metabolism (Appendix A).

Pyroptosis is triggered by innate immunity perturbations [36,37]. Variant cues, such as viral and bacterial products (LPS, is one of the major ones), trigger the formation of inflammasome followed by the activation of specific caspases including CASP1, CASP4, and CASP5. Such caspases cleave substrates such as GSDMD leading to pyroptosis. Moreover, pro-inflammatory interleukins IL1Beta and IL18 are also often secreted (Appendix A).

Ferroptosis relies on derailed redox regulation [38,39]. When antioxidant enzymes, mainly GPX4, are overwhelmed, membrane lipids are oxidized to peroxides, causing fatal alterations in membrane permeability, calcium waves, changes in membrane potential, and cell death (Appendix A).

### 2.4. The Interplay between Modes of Cell Death and Switch Points

From the detailed RCD network and the RCD modules, it is possible to deduce a scheme of the major players and interplay among them (Appendix A, Figure 3). For simplification and clarity of the message, the signaling is represented in the form of an activity flow diagram indicating either activation or inhibition. The inputs at the top of the initiation layer dictate the choice of the pathway. However, the regulation of the chosen pathway is not trivial, because the same regulator can influence different cell death modes, depending on the presence of activators and inhibitors. It is also important to note that the formation of high molecular weight complexes is an essential step for the initiation of all modes of cell death. Moreover, cooperation between cell death modes can occur. Once the choice of the cell death mode is made, the signaling propagates to the execution layer, resulting in different forms of cell death (outputs), each of which has a particular phenotype (Figure 3).

Engagement between the ligands and death receptors results in the initiation of signaling, as after interaction of TNF with its receptor, RIPK1 and CFLAR are recruited to the death receptor signaling complex (DISC) thanks to TRADD along with CASP8. Three major pathways emerge, depending on the recruited players and the degree of CASP8 oligomerization: apoptosis, necroptosis and pro-survival signaling. CFLAR prevents apoptosis triggering. In this situation, recruited RIPK1 is activated through polyubiquitination. Polyubiquitinated RIPK1 then activates IKK and MAP3K7 leading to the activation of the pro-survival transcription factor NF-κB. Moreover, RIPK1 phosphorylation by IKK and MAP3K7 prevents its interaction with FADD and CASP8 thus favoring RIPK1-independent apoptosis [40]. Members of the inhibitor of apoptosis protein family (IAP) also play an important role in these interactions. For example, one IAP called XIAP constitutively inhibits caspases in the physiological state. Other IAPS like BIRC2 and BIRC3 lead to an upregulation of anti-apoptotic factors and also ubiquitinate RIPK1, thus, triggering pro-survival NF-κB signaling [41,42]. However, some IAPs provoke CYLD dependent RIPK1 de-ubiquitination, that results in RIPK1 release from the necrosome complex, which drives extrinsic apoptosis [43].

Apoptosis strongly depends on energy supply, therefore in the absence of ATP, cells tend to undergo necrotic rather than apoptotic death. In specific circumstances, when AMPK is activated, BECN1 may favor CASP8 cleavage and apoptosis activation. Interestingly, both apoptotic and necroptotic machinery can be co-activated, if a combination of stimuli is applied on the cell or when apoptotic signaling is activated while mitochondria become permeabilized and releases factors that favor necroptosis, such as CYPD [44]. Autophagy normally maintains the energy status in the cell [34,45], however prolonged autophagy may result in cellular demise [46]. 

Reactive oxidative species (ROS) are widely implicated in most RCD modes. As they can act both as initiator and executor, depending on the RCD subroutine, ROS tend to increase interactions between cell death pathways. ROS accumulation favors BAX/BAK recruitment to mitochondria and thus acts as a major initiator of intrinsic apoptosis. ROS is also essential for the successful release of Cytochrome C (CYCS) from mitochondria. In the same way, parthanatos can be a consequence of prolonged oxidative stress. Although the precise mechanism remains unknown, caspase activation has also been shown to generate ROS via the mitochondrial electron-transport chain [47]. Likewise, ferroptosis is based on the generation of an excessive amount of ROS.

### 2.5. Access and Navigation of RCD Map

The RCD map can be browsed online at https://navicell.curie.fr/pages/maps_rcd.html. The map is presented in three independent platforms, namely NaviCell, MINERVA [25] and NDEx [26] that can be accessed from the map home page. The map components are clickable, making it interactive. The extended annotations of the map components contain a rich tagging system converted to links. This allows tracing the involvement of molecules into different map sub-structures as meta-modules and modules. The tagging system also allows using the map as a source of annotated signatures of different cell death modes (Figure 2 and Appendix A).

The semantic zooming feature of NaviCell [48] simplifies navigation throughout the large collection of molecular interactions, revealing a readable amount of details at each zoom level. A gradual exclusion of details allows the exploration of map contents, going from the detailed towards the top-level view. The hierarchical structure of the RCD map as described above facilitated the generation of several zoom levels for web-based navigation of the map (Figure 2).

### 2.6. Comparison of RCD Map with Similar Pathway Resources

Number of regulated cell death mechanisms are described in several existing pathways resources as KEGG [14] and REACTOME [15] pathway databases.

There are additional cell death-related resources in the field, among others there is DeathBase (http://www.deathbase.org), a database on proteins involved in cell death [49] and ApoptoProteomics (http://apoptoproteomics.uio.no), an integrated database for analysis of proteomics data which also contains valuable information matching cell death proteins to anti-cancer drugs [50]. Finally, there is a specific yeast programmed cell death online platform (http://ycelldeath.com/index.html) that collects various type of information of three types of PCD in yeast, Apoptosis [51], Necrosis, and Autophagy. The cell death-type specific pathway repositories are rare. So far there is only the Autophagy Regulatory Network database that gathers information on protein-protein interactions related to Autophagy process and also connects these components and regulators with previously existing signaling pathways in public resources [52]. To our knowledge, there are no other databases specific to other cell death processes. 

Majority of abovementioned resources do not represent cell death processes in a form of process description diagrams. In fact, the closest to the RCD map representation mode and therefore the most comparable databases are KEGG and REACTOME, where pathways are shown using a syntax similar to one applied to generate the RCD map. The RCD map has been compared to the relevant pathways describing cell death-related processes from the KEGG and REACTOME pathway databases. Altogether these relevant selected pathways cover 956 HUGO names in KEGG and 805 HUGO names in REACTOME. These lists were compared with the RCD map that contains 891 HUGO names (Appendix A). This number is different than the one reported in Table 1 as we choose to only select HGNC approved gene symbol to have a reliable comparison with the two other databases. For example, “cleaved caspase1*” and “caspase 1*” count as two different entities in Table 1 but correspond to only one HGNC approved symbol (CASP1) in Appendix A. 

Some RCD modules represent the processes at a more detailed level in terms of the number of gene names, comparing to the corresponding pathways from KEGG and REACTOME databases. For example, Programmed Cell Death pathway in REACTOME contains 178 HUGO names, this corresponds to the Apoptosis and Necroptosis modules in RCD map, together covering 284 HUGO names. On the contrary, the necroptosis module is slightly more enriched in the KEGG pathway than in the RCD map (165 in KEGG vs. 154 in ACSN). However, in KEGG, the parthanatos pathway (especially the groups of histone genes) is included within the necroptosis module. In contrary, in the RCD map, parthanatos is described as an independent module and is updated with the most recent publications related to cancer. Some metabolic pathways are more represented in other databases, comparing to the RCD map, as this applies to the pentose phosphate pathway and porphyrin metabolism (that actually includes chlorophyll metabolism) in KEGG, as well as to fatty acid metabolism in REACTOME (Appendix A). However, it is important to stress that metabolic reactions are not within the scope of the RCD map unless they modulate cell death-related processes. 

The overlap of the RCD map HUGO names with other pathways reaches 47% for KEGG and 36% for REACTOME. 323 out of 805 HUGO names from REACTOME pathways and 423 out of 956 proteins from KEGG pathway are also present in the RCD map (Figure 4A and Appendix A).

Remarkably, 375 HUGO names are unique for the RCD map (Figure 4A and Appendix A). These unique entities are homogeneously distributed across the map. More recently described RCD modes, such as ferroptosis and pyroptosis that are carefully represented in the RCD map are close-to-completely absent from KEGG and REACTOME. Altogether, we conclude that the contents of the RCD map are not redundant with other pathway databases and that a significant proportion of entities are unique to the RCD map. 

We also compared the publications used to annotate the RCD map and the corresponding selected pathways in KEGG and REACTOME resources. Of note, 815 papers out of 829 papers that were used to annotate the RCD map are unique and are not referenced in the annotations of selected pathways from KEGG or Reactome (Appendix A). Although the median age of the literature references in the RCD map is older than in the corresponding pathways from the other two databases, 25% of the papers referenced in the RCD map have been published relatively recently, between 2010–2018 (Figure 4B). 

Finally, the journal types represented in the three databases were compared. The range of the journals used for annotation of the RCD map and the other two databases is relatively similar. However, the distribution of papers from different types of journals is rather different. In the annotations of the RCD map, the citations to journals in the area of molecular biology are overrepresented, compared to the other two databases. This indicates that the RCD map dwells into the molecular mechanisms rather than the phenotypic description of the process (Figure 4C,D). 

Taken together, the results of database comparisons indicate that the RCD resource is topic-specific, and covers all known modes of cell death mechanisms and their interplay. The thoughtful layout and visual organization of the biological knowledge displayed in the map make it an attractive resource for data analysis.

### 2.7. Application of the RCD Map to Neurodegenerative and Malignant Diseases

Alzheimer’s disease (AD) is the most prevalent neurodegenerative disease in which neurons progressively succumb [53]. Nevertheless, the precise molecular processes resulting in neuronal cell death are elusive. To explore the RCD mechanisms involved in AD, gene expression data from three different studies were analyzed (see Materials and Methods), focusing on the Hippocampus, which is the most affected brain area in this disease [54]. In addition, the transcriptome of three non-small cell lung cancer (NSCLC) cohorts were studied, knowing that cancer is a disease characterized by the unwarranted survival of tumor cells. 

The pathway scoring method ROMA was applied to quantify the module activity scores for both diseases using the module definition from RCD map as detailed in Martignetti et al. [55]. The module activity values were plotted on the RCD map using the BiNoM plugin of Cytoscape to visualize the differences between the two diseases [56]. 

The ROMA scores of AD and NSCLC exhibited rather an inverse trend (Figure 5 and Appendix A). Regarding RCD types, the modules corresponding to the TRAIL response, Pyroptosis, and Dependence Receptors were more active in AD (Figure 5A). In contrast, in NSCLC, the ligand-receptor modules (TNF response, TRAIL response, and FAS response), as well as some modules of the signaling layer (Pyroptosis and Dependence Receptors), are less active (Figure 5A, Appendix A).

Of note, most modules corresponding to the pyroptosis module appeared to be more active in Alzheimer disease. In addition, in NSCLC several metabolism-related modules (including glucose metabolism, oxidative phosphorylation and the citrate cycle), as well as ER stress, were more active (Figure 5B,C). This metabolism-related modules activity confirm previous studies [57,58], including on the difference between Alzheimer’s disease and lung cancer [55]. Indeed, the integrated comparison of the RCD map across AD and NSCLC is in line with speculations on the inverse comorbidity between both diseases [59,60,61], as well epidemiological studies suggesting that NSCLC occurs less frequently in AD patients than in age-matched individual without AD [62]. 

In the next step, the top contributing genes for each disease were identified. For this, the correlation coefficients in all the studies were calculated and those genes that had a correlation coefficient of minimum 0.5 (absolute value), with a significant p-value and appeared in at least 2 of the 3 data sets were selected (Table 2). As a result, a list of genes that contribute either positively or negatively to each module was retrieved. 

In the case of Alzheimer, pyroptosis has been suggested and reviewed as a CASP1-dependent response to chronic aseptic inflammation [37,63]. Experimental evidence has linked pyroptosis in Alzheimer to the NLRP1 inflammasome [64] or NLRP3 inflammasome [65]. However, most studies correlating Alzheimer and pyroptosis have been performed in rodent models. Here, we identified *IL18*, *CASP4*, *GBP2*, *CASP1*, and *AIM2* as the genes that were contributing most to the pyroptosis module. *IL18* gene is over-expressed in brains of AD patients [66]. *CASP4* expression has been hypothesized to mediate inflammatory responses in AD pathology [67]. *CASP1*, together with other genes encoding caspases, is overexpressed in AD patients [68]. *AIM2* has been found in mouse models to promote IL1B secretion by neurons, which might also participate in AD pathology [69]. However, *GBP2* expression has not yet been evaluated for its potential role in AD pathology. Recently, Saresella and colleagues found that inflammasome components (NLRP1, NLRP3, PYCARD, CASP1, CASP5, and CASP8) and downstream effectors (IL1B, IL18) were upregulated in peripheral blood mononuclear cells from patients with moderate and severe AD [70]. All these findings support our results using the RCD map; nevertheless, future research is needed to elucidate if neuroinflammation leads to pyroptosis during AD pathology. 

In cancer, ER stress has been identified as an adaptive response that favors either growth or apoptosis [70,71]. In addition, ER stress has also been related to chemotherapy resistance [72]. In our study, we identified the genes *PDIA6*, *PDIA4*, *DNAJB11*, *SEC61G*, *SEC61A1,* and *CREB3L4* as positively contributing genes and *ITPR1*, *RYR2*, *NFKB1,* and *NLRC4*, as negatively contributing genes for the ER stress module. *PDIA6* and *PDIA4* have been demonstrated to be overexpressed in NSCLC biopsies resistant to chemotherapy with cisplatin, and their silencing actually may reverse drug resistance [73]. The *NFKB1* gene has been described to be a key player in the ER stress pathway and cancer survival mechanisms [74,75]. *NLRC4* is downregulated in lung cancer cases [76]. *NLRC4* contains a caspase recruitment domain (CARD) through which it can regulate apoptosis via NF-κB signaling pathways [76], suggesting a possible link between *NLRC4* and *NFKB1* genes in this module. In contrast, there are no consistent reports on the possible involvement of *DNAJB11*, *SEC61G*, *SEC61A*, and *CREB3L4* in NSCLC. 

Finally, we performed the enrichment study using the Gene Set Enrichment Analysis (GSEA) to complement the ROMA analysis. We observed the same pattern of processes involvement in AD and LC. In particular, metabolism and mitochondria activity-related modules are activated in the LC comparing to AD. In contrary, the modules responsible for the initiation of RCD are inhibited in LC vs. AD Appendix A and Appendix A). 

### 2.8. RCD Signatures in Different Ovarian Cancer Subtypes 

Ovarian cancer, the second most prevalent of the female genital tract, with a high mortality rate [77,78] (https://seer.cancer.gov/statfacts/html/ovary.html), encompasses a heterogeneous group of tumors because different genetic, morphological and pathological characteristics are involved in the disease initiation and progression^91,92^. We analyzed transcriptomics and copy number data from a cohort of 489 high-grade serous ovarian cancer accessible via The Cancer Genome Atlas (TCGA) [79,80]. Using a non-negative matrix factorization approach, these tumors have been clustered into four subtypes, namely: differentiated, immunoreactive, mesenchymal, and proliferative, as reported elsewhere [80]. 

Relying on these subtypes definitions, we applied ROMA analysis (see above) to mRNA expression data to this data set, while comparing the four ovarian cancer subtypes. The visualization of the ROMA scores in the context of RCD map revealed important differences among these subcategories (Figure 6). The differentiated subtype exhibited several active modules in the area of metabolism (mitochondrial metabolism, glucose metabolism, oxidative phosphorylation and TCA cycle). In contrast, apoptosis-related and starvation/autophagy-related modules appeared to be deactivated (Figure 6A). As to be expected [81], the immunoreactive subtype was characterized by the upregulation of the majority of the modules related to death receptor pathways and caspase-related executors (Figure 6B). In addition, we observed the activation of the modules initiated by inflammation, such as pyroptosis and—to a lower degree—starvation/autophagy (Figure 6C). The mesenchymal type, being the most aggressive one, displayed high activity in some death-executing modules, with a notable suppression of oxidative phosphorylation and TCA cycle as well as the antioxidant response (Figure 6B). The proliferative subtype exhibited higher activity of the modules related to fatty acid biosynthesis, glutamine metabolism, and nuclear integrity (including the DNA damage response), all in line with the high replicative activity of this cancer subtype. In contrast, the majority of functional modules responsible for regulation and execution of cell death were strongly downregulated in the proliferative subtype (Figure 6E).

Visualizing the copy number (CN) data of the different subtypes of ovarian cancer in the context of RCD map allowed us to see that the proliferative type corresponds to a higher number of CN gains, indicating massive genomic aberrations, compared to other subtypes (Figure 6), that represents an important negative prognostic factor [80,82]. For example, it is now known that *YWHAZ* promotes ovarian cancer metastasis by modulating glycolysis [83]. CN variations in *WISP1* have been associated with endometrial adenocarcinoma [84] whereas *NDRG1* has been reported as a tumor suppressor in ovarian carcinogenesis across distinct ovarian subtypes [85,86] (Appendix A), therefore it is thus not surprising to find these genes as top hits across ovarian cancer subtypes.

In synthesis, data visualization via RCD maps may constitute a useful tool for determining functionally relevant differences among cancer subtypes and between human disorders. 

## 3. Discussion

Different RCD modes share multiple common molecules across inputs, signaling and execution players. There is often a set of cues, such as cellular energy status, external signals, as well as type to DNA damage, that, in combination, will dictate a choice of cell death scenarios. How this coordination is done is to a large extent poorly understood. The existing linear and disconnected representations of regulated cell death mechanisms are far from satisfactory. In this work, we attempted to gather all available information about RCD mechanisms and to represent them in a structured and computer-readable manner. The comprehensive RCD map covers the initiating phases, the signaling phase during which the mode of the cell death is chosen, as well as the final and fatal execution phase of the process.

Understanding the interplay between RCD modes represented in the form of a comprehensive signaling map can help to evaluate the status of cell death modes in human diseases. This may be critical for choosing the correct treatment strategy and predicting therapeutic responses. For example, restoring caspase activity in cancer treatment by selective induction of apoptosis is one of the common goals of antineoplastic treatments, though hampered by the activation of apoptosis-inhibitory pathways. For this reason, it has been suggested to develop alternative agents that would activate alternative routes of RCD. As a speculative scenario, the induction of necroptosis, ferroptosis, or autophagy-dependent death might offer an opportunity to kill cancer cells and this way to restore their sensitivity to cytotoxic drugs [11,16,33,34]. The RCD map will help to select the alternative routes by for each particular case. 

Beyond this possibility, we demonstrate here that this resource can be applied to other pathologies. As aberrations in cell death control are involved in numerous diseases, the RCD map might be taken advantage of to explore pathologies including infectious diseases and degenerative processes. We demonstrated in this work that the RCD map is useful for the comparison of Alzheimer disease, a state of excessive cell death, and NSCLC, a pathology linked to the suppression of cell death pathways.

In general, one of the obvious and immediate application of the RCD maps is identification of network-based signatures of diseases with respect to the cell death mechanisms status. Indeed, as we showed in our application examples, integration and analysis of data in the context of the map provides a glimpse not only on the deregulated RCD processes in a studies disease or sample, but also indicates the most contributing players, which can be future therapeutic targets. The second way of RCD map applications is modelling the switches between the cell death modes. Depending on the modelling approach, the comprehensive or the simplified chart of the RCD network can be used. 

## 4. Materials and Methods

### 4.1. Map Availability

The RCD map is freely available on the web page https://navicell.curie.fr/pages/maps_rcd.html. The map is provided in three platforms, NaviCell, MINERVA and integrated into the repository NDEx. The RCD map can be downloaded in several exchange formats. In addition, the content of the RCD map grouped per modules is available and downloadable in the form of GMT files suitable for further functional data analysis. 

### 4.2. Map Construction 

The signaling network of RCD regulation was manually constructed and based on information extracted from literature curation retrieved from the PubMed database. The map was composed using the CellDesigner tool [35] and the standard visual syntax Systems Biology Graphical Notation (SBGN) [36] for representing molecular biology mechanisms. The software creates a structured representation in Systems Biology Markup Language (SBML), suitable for further computational analysis [37]. Each entity of the map is annotated by corresponding references and specific notes. 

### 4.3. Data Model

The RCD map was constructed using a methodology previously described [22,27]. The maps are drawn using Process Description (PD) language of Systems Biology Graphical Notation (SBGN) syntax. The CellDesigner format represents a proprietary extension of the Systems Biology Markup Language (SBML) [28]. The most basic unit of RCD map is a biochemical reaction, formed by reactants, products and various types of regulators (Figure 1 and Appendix A). The data model includes the following molecular objects: proteins, genes, RNAs, antisense RNAs, simple molecules, ions, drugs, phenotypes, complexes. These objects can play a role of reactants, products and regulators in a connected reaction network. The objects “phenotypes” play a role in biological process outcome or readout. Edges on the maps represent relations between biochemical reactions and chemical species or reaction regulations of various types. Different reaction types represent posttranslational modifications, translation, transcription, complex formation or dissociation, transport, degradation, and so on. Reaction regulations include catalysis, inhibition, modulation, trigger, and physical stimulation. The naming system of the maps is based on HGNC-approved gene symbol, named as ‘HUGO identifiers’ in the text, for genes, proteins, RNAs and antisense RNAs and CAS/ChEBI identifiers for drugs, small molecules and ions. In addition, in some cases, individual or generic entities are named by commonly used synonyms. In this case, the name is followed by an asterisk (e.g., ‘p53*’ instead of ‘TP53’). In all cases, the HUGO identifier is provided in the protein notes. Subcellular localization of molecular entities is an important factor for the functioning and activity of the signaling processes. Cellular compartments such as cytoplasm, nucleus, and mitochondria recapitulate the cellular architecture on the map (Appendix A). 

### 4.4. Literature Selection Rules

The rules for literature selection are as following: First five to ten review articles in the field were chosen for extracting the consensus pathways and regulations accepted in the field and for approaching to the literature suggested in these reviews. This allowed to construct the “backbone” of the network and to define the biological processes boundaries (Figure 1). Further, information from original papers, preferably most recent studies, is analyzed and added to enlarge and enrich the network. In the majority of cases, the decision about adding a biochemical event or regulation or a process to the map has to be supported by at least two independent studies performed in different teams. 

### 4.5. Map Annotation Format

Each entity and reaction on the map is annotated using the NaviCell annotation format (Appendix A). The annotation includes three sections namely “Identifiers”, “Maps_Modules” and “References”. “Identifiers” section provides links to the corresponding entity descriptions in HGNC, UniProt, Entrez, GeneCards, REACTOME, KEGG, and Wikipedia databases. “Maps_Modules” section includes links to modules where the entity is found. “References” section includes notes added by the map manager and links to relevant publications in PubMed. Annotations can be associated with a molecular entity (such as protein) as well as to its particular modification (such as a particular post-translational modification). NaviCell (described below) provides mechanisms of cross-referencing RCD map with other molecular and pathway databases, such as KEGG PATHWAY, Reactome, Atlas of Genetics and Cytogenetics in Oncology and Haematology, GeneCards, and Wiki Genes.

To evaluate the reliability of the mechanisms represented on the map, each reaction is provided with a confidence score as described elsewhere [22]. There are two values indicating the publication score (REF) and the functional proximity score (FUNC) that is computed based on the Human Protein Reference Database protein-protein interactions (PPI) network.

### 4.6. Module Tagging 

A systematic tagging system was applied for annotation of each entity with module tags. The tags are converted into the links by the NaviCell factory in the process of online map version generation. The links allow to trace participation of entities in different modules of the map and also facilitate shuttling between these structures.

### 4.7. Generation of NaviCell Map with NaviCell Factory 

CellDesigner map annotated in the NaviCell format is converted into the NaviCell web-based front-end, which is a set of html pages with integrated JavaScript code that can be launched in a web browser for online use. HUGO identifiers in the annotation form allow using NaviCell tool for visualization of omics data. A detailed guide of using the NaviCell factory embedded in the BiNoM Cytoscape plugin [70] is provided at https://navicell.curie.fr/doc/NaviCellMapperAdminGuide.pdf.

### 4.8. Map Navigation

The content of the RCD map is provided in the form of an interactive global map. The visualization and exploration of the RCD map are empowered by the NaviCell web-based environment (https://navicell.curie.fr). NaviCell uses Google Maps engine and principles for navigating through the content of the map. Thus, the logic of navigation as scrolling, zooming, and features as markers, callout windows and zoom bar are adopted from the Google Maps interface. Map queries can be done using the search window or by selecting the entities of interest in the selection panel. The development of semantic zooming in NaviCell simplifies navigation through large maps of molecular interactions by providing several levels of the map view. To facilitate the exploration, each level of zoom on the map exposes a certain depth of detail. In addition, navigation is also simplified by the modular structure of the map where each module map can be visualized individually. Shuttling between the global map to modular maps using the map icon (globe) (Figure 2 and Appendix A).

### 4.9. Web-Based Platforms Support 

The RCD map is available at other platforms such as MINERVA [25] and NDEx [26]. To integrate maps within NDEx, CellDesigner map was first loaded to Cytoscape using the BiNoM Cytoscape plugin and then uploaded on NDEx using the CyNDEx Cytoscape plugin.

### 4.10. Map Structural Analysis

The map backbone structure was manually built based on a selection of major inputs and output in each functional module of the map (Appendix A). For each module, major inputs and outputs were selected on the map by tags from the RCD map, extracted and further manually refined based on a number of interactions and reviews of the literature. The backbone structure of the network was restored using the information from the comprehensive RCD map. Each cell death mode was first represented as a separate diagram (Appendix A). All diagrams were gathered together to create the global scheme, where interplays between different cell death modes were included (Figure 3).

### 4.11. Data Acquisition and Groups Definition

Gene expression datasets from lung cancer and Alzheimer’s disease (GSE36980, GSE48350, GSE5281, GSE19188, GSE19804, GSE33532) were downloaded from the Gene Expression Omnibus (GEO, https://www.ncbi.nlm.nih.gov/geo/). For the testing, we grouped the cases of each disease and compared them to their respective controls. The ovarian cancer data was obtained from The Cancer Genome Atlas (TCGA) [80,81]. The group definitions for the ovarian cancer cases were used as previously described by using a non-negative matrix factorization approach [80]. 

### 4.12. Module Scores Calculation and Visualization

The Representation of Module Activity (ROMA) analysis [55] was performed using the “rRoma” and “rRomaDash” R packages (available at https://github.com/sysbio-curie/rRoma and https://github.com/sysbio-curie/rRomaDash). ROMA quantifies the activity of a group of genes biologically related, referred to as a module in the RCD map (see section” Structure of the Regulated Cell Death map”), by calculating the largest amount of one-dimensional variance in the expression of the genes that constitute a module across the samples. With the analysis provided by ROMA, we identified the over-dispersed modules in RCD map. Further, one-tailed pairwise t-tests using the ROMA scores from all modules was done, to determine module activity directionality. Then the calculation of the top contributing genes was performed per module using Pearson’s correlation, selecting those with an absolute correlation coefficient of 0.5 minimum. The association was considered significant when *p* < 0.01. Colored map images were obtained using the function “Stain CellDesigner map” from BiNoM Cytoscape plugin [56] using XML map files and the average relative ROMA scores from the analysis described above. In addition, Gene Set Enrichment Analysis (GSEA) [87] was also performed to complement the results obtained by the ROMA Analysis for lung cancer and Alzheimer’s disease (Appendix A and Appendix A). 

## 5. Conlusions

In summary, we anticipate that the RCD map will become useful for bioinformaticians and translational scientists interested in exploring aberrations in the cell death pathways across distinct human pathologies. 

## Figures and Tables

**Figure 1 cancers-12-00990-f001:**
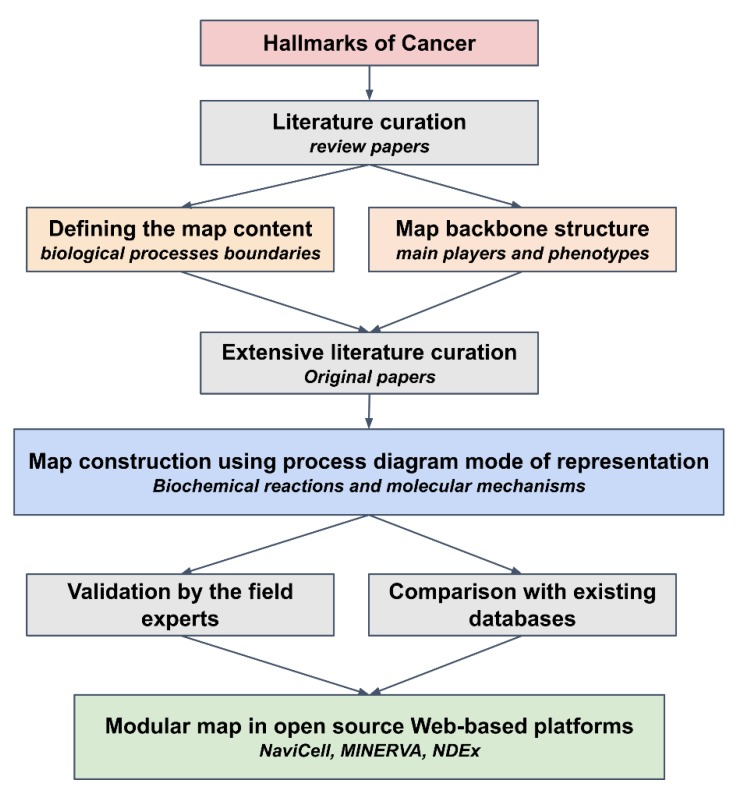
Regulated cell death map construction workflow. The scheme depicts the steps of map construction starting from a collection of cancer-specific and regulated cell death (RCD)-related information about individual molecular interactions from scientific publications and databases, manual annotation and curation of this information, then organization of the formalized knowledge in form of a global map with a modular structure.

**Figure 2 cancers-12-00990-f002:**
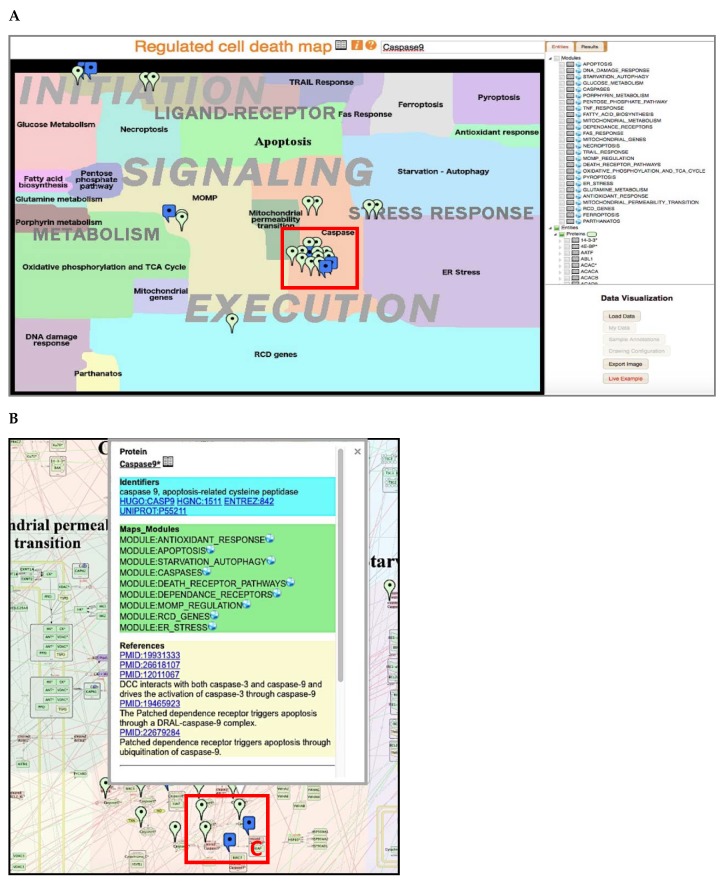
Regulated cell death map browsing, zooms, and entities annotations. (**A**) Map interface in NaviCell-powered Google Maps platform with a top view layout of RCD map. The interface includes the map window, selection panel, data analysis panel and upper panel. Map querying is possible via the search window or by checking on the entity in a list of entities in the selection panel that will drop markers all over the map (for example, Caspase 9). (**B**) Zoom of a fragment of the map and callout window. Clicking on a marker opens a callout window containing three sections: “Identifiers” with links to external databases; “Map Modules” with links to functional modules where the entity of interest is found, “References” with links to PubMed, and annotation notes. Clicking on the “globe” icon opens the corresponding map. Clicking on a “book” icon opens a detailed annotation page. (**C**) Zoom of a fragment of a module showing the most detailed level of the representation of molecular mechanisms. The participation of the selected molecule (e.g., caspase 9) in various reactions and complexes can be seen.

**Figure 3 cancers-12-00990-f003:**
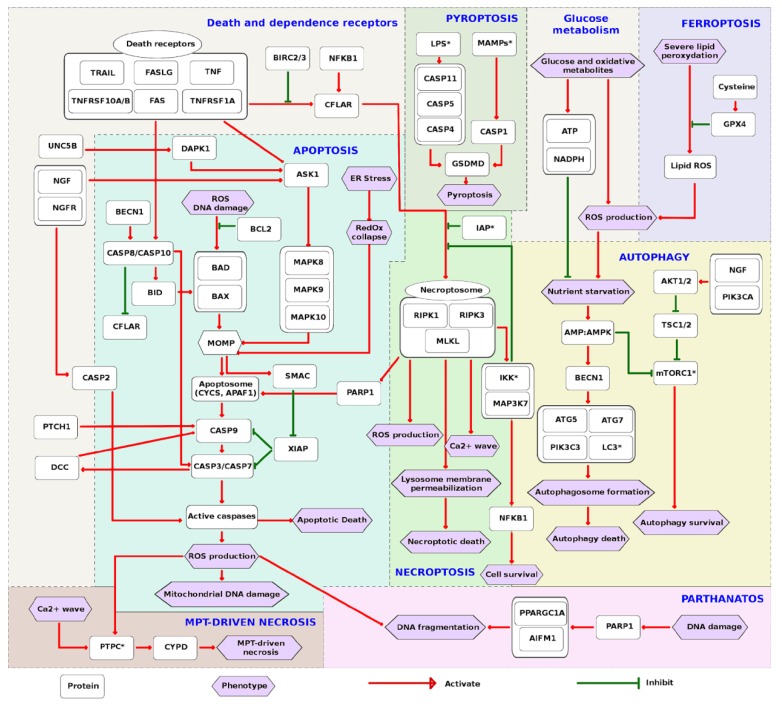
The interplay among regulated cell death modes. Input molecules, major players, and resulting phenotypes were selected for each RCD map module. The choice was based on the number of interactions on the map and information from the literature. Each module map was manually pruned, while keeping intact hub players, functionally important molecules and their corresponding interactions. The resulting graphs were manually integrated together and the interaction between cell death modes was added. The diagram was converted into an activity flow diagram by replacing reaction edges by influence edges, either activation or inhibition, in order to achieve the final layout. Some molecular entities are grouped into a generic entity representing either a molecular complex or a reaction cascade. HGNC nomenclature for molecules naming was used, except for entities marked by stars, for which common names were used. RCD modes are highlighted by the colored semi-transparent background. Most of the pathways are drawn from the top (initiation) to the bottom of the figure (execution). Some incidences of cross-talk between pathways are clearly illustrated. ROS production is a crucial input and output of RCD. Death receptor can trigger a great variety of responses, even cell survival (via induction of IAP, NFKB1). Apoptosis, necroptosis and autophagy are interconnected processes (through multifunctional implications of CFLAR, PARP1, IAP, BECN1). Energy supply is a major regulator of cell fate. IAP: Inhibitors of apoptosis proteins; LPS: lipopolysaccharides; MAMPs: microbe-associated molecular patterns; PTPC: permeability transition complex; MPT: mitochondrial permeability transition; LC3: microtubule-associated protein 1 light chain 3 beta (MAP1LC3B).

**Figure 4 cancers-12-00990-f004:**
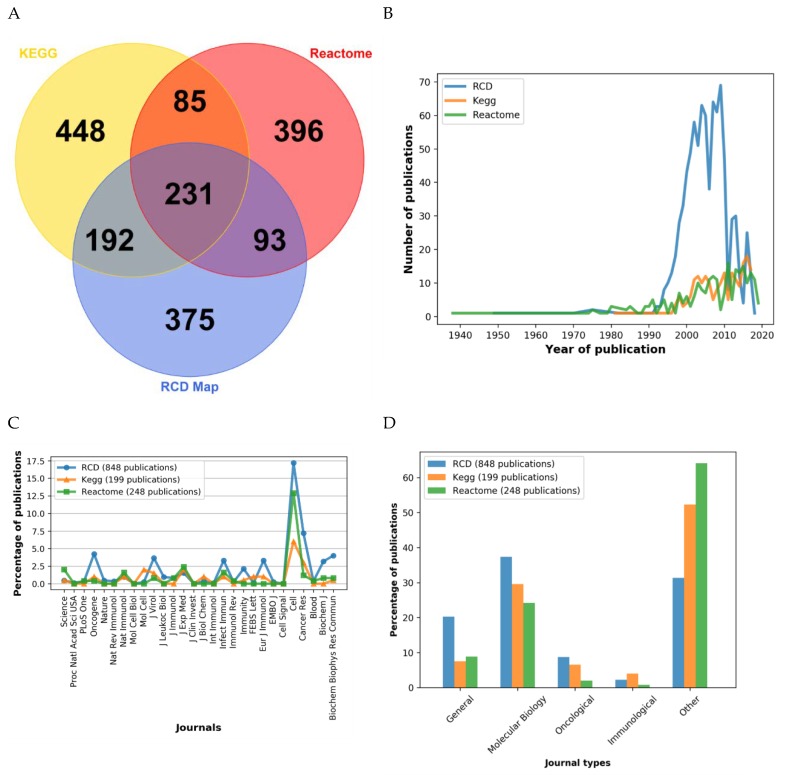
Comparison of RCD Map with other pathway resources. (**A**) Comparison of HUGO names content in selected pathways from REACTOME and KEGG databases with the RCD map. The Venn diagram shows the overlap of HUGO names between different databases. Distribution of (**B**) publication years and (**C**) journals annotating the selected pathways from the REACTOME, KEGG, and RCD map. (**D**) Relative distribution of journals types used for annotation of the selected pathways from REACTOME, KEGG, and RCD map.

**Figure 5 cancers-12-00990-f005:**
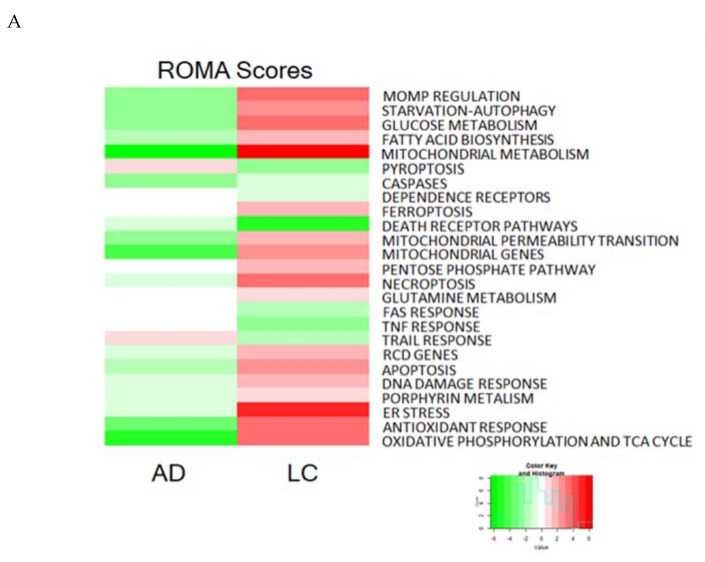
Visualization of average ROMA modules activity scores using expression data from Alzheimer disease (AD) hippocampus samples and non-small cell lung cancer (NSCLC) specimens in the RCD map. (**A**) Heatmap representing ROMA scores for the two diseases (each respective to its normal controls). Staining of RCD map with ROMA scores from (**B**) AD data and (**C**) NSCLC data. The plotted values correspond to the relative ROMA module score compared to controls (as in A). Top contributing genes are represented in their locations on the map, purple positively contributing genes and yellow negatively contributing genes.

**Figure 6 cancers-12-00990-f006:**
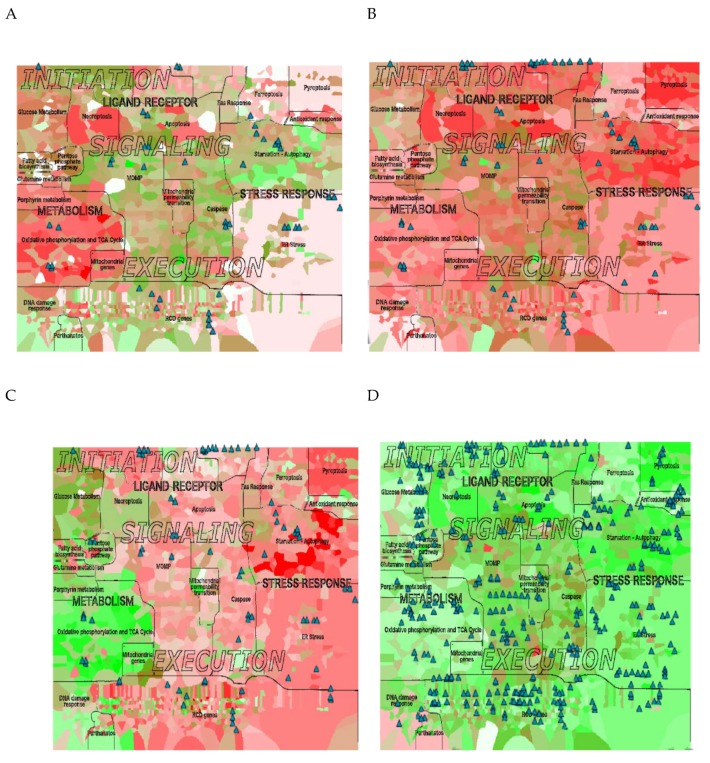
Visualization of expression and genomic data from ovarian cancer groups. (**A**) differentiated, (**B**) mesenchymal, (**C**) immunoreactive, (**D**) proliferative subtype. The background color represents the expression value per protein (red: up-regulated, green: down-regulated). The glyphs (triangles) represent gene copy number gains above 4 copies. (**E**) Heat map of the ROMA scores per subtype.

**Table 1 cancers-12-00990-t001:** Structure and content of the RCD map.

Layer/Module	Chemical Species (Entities)	Proteins	Genes	RNAs	asRNAs	Reactions
Initiation (reversible)						
Stress response						
Antioxidant response	213	151	4	9	12	95
DNA damage response	128	59	3	3	5	80
ER stress	382	161	45	44	4	291
Starvation-autophagy	438	165	19	31	31	290
Ligand receptor						
Death receptor pathways	640	287	23	27	32	469
Trail response	36	25	0	0	0	18
Fas response	29	24	0	0	0	14
TNF response	87	39	3	3	0	57
Dependence receptors	78	52	0	0	0	36
metabolism						
Fatty acid biosynthesis	78	34	1	3	2	33
Glucose metabolism	190	112	0	0	0	104
Glutamine metabolism	42	20	0	0	0	20
Pentose phosphate pathway	64	12	2	2	2	39
Porphyrin metabolism	51	17	0	0	0	26
Mitochondrial metabolism	608	349	1	35	0	360
Oxidative phosphorylation and TCA cycle	232	179	0	0	0	109
Mitochondrial genes	114	116	2	30	0	36
Signalling (rewirable)						
Apoptosis	584	246	34	40	57	401
Necroptosis	242	118	0	0	0	188
Ferroptosis	193	65	17	19	0	110
Parthanatos	80	17	1	0	0	64
Pyroptosis	112	36	1	0	0	79
Execution (irreversible)						
MOMP regulation	630	319	19	48	17	396
Mitochondrial permeability transition	57	36	0	0	0	33
Caspases	318	158	5	5	12	221
RCD genes	633	135	168	179	90	365
RCD global map	2657	1008	215	260	93	2020

The map gathers 2657 entities including over 1000 chemical species as proteins, genes, RNA, metabolites, ions, small molecules as drugs. Numbers refer to unique component counts/functional module of the RCD map. Zero values indicate absence of the molecule category in the corresponding functional modules of the RCD map.

**Table 2 cancers-12-00990-t002:** Top contributing genes per module for Alzheimer and lung cancer datasets.

Module	Top Contributing Genes AD	Top Contributing Genes LC
MOMP regulation	*ATG5* *, VDAC3, CUL2, CYCS, HK1, MAPK6, BMF, RPS6KA1, MAPK10, VDAC2*	*DIABLO* *,* *RPS6KA1* *, CISH, CSNK2A1, C1QBP, FOXO1, PPP1CC, BCL2L11, SLC25A5, VDAC2 *
Starvation Autophagy	*CAB39* *, ATG5, MAPK6, MAPK8, ACACA, PINK1, RPS6KA1, CASP8, MTOR, ATG12*	*PARP1* *, SESN1, LDHA, BCL2L11, PI3KC3, RPS6KA1, PINK1, ATG3, ATG12*
Glucose Metabolism	*PFKM* *, PRKACB, PRKAA2, LDHA, NOTCH1, H6PD, ALDOA *	*PGK1* *, LDHA, ANAPC11M ALDOA, LDHC, PKM, ANAPC2*
Fatty acid biosynthesis	*PRKAA2*, *PPAT*, *CAB39*, *PRPS1*, *PRKACB*, *ACACA*	*ACLY*, *ACSS2*, *MLST8*, *PRPS1*, *PRPS2*
Mitochondrial metabolism	*SLC25A14* *, NDUFA9, FH, DLD, NDUFAF1, HK3, TNFRSF1A, ALOX15B*	UQCRH, ATP5G1, SLC25A10, CYC1, TACO1, NDUFAB1, PFKB2, RIPK3, SESN1, PLA2G4C
Pyroptosis	*IL18*, *CASP4*, *GBP2*, *CASP1*, *AIM2*	*CASP1*, *GBP2*, *IL18*, *CASP4*, *IL1B*
Caspases	*YWHAG*, *MAPK8*, *MAPK6*, *CAB39*, *LMNA*, *HIP1*, *RIPK3*, *MTOR*	*G0S2*, *BIRC3*, *PIDD*, *FOXO1*, *RNF41*
Dependence receptors	*AATF* *, APPL2, PPP2CB, NGF, CARD8, NSMAF*	*PIK3CA*, *NEDD4*, *TRAF6*, *PPP2CB*, *NSMAF*
Ferroptosis	*NFE2L2*, *NQO1*, *FTL*, *ACSF2*, *SLC3A2*, *FTH1*	*GCLC*, *GCLM*, *GSR*, *MAFG*, *GSS*, *SLC3A2*
Death receptor pathways	*B4GALT6*, *GUCY1B3*, *CUL3*, *RBX1*, *RIPK3*, *HIF1A*, *NDUT5*, *CASP3*	*CYLD* *, PARP1, BRCA1, H2AFX, PRKDC, CARD8, STAT5A, FOXO1, MAGED1, TNFAIP3*
Mitochondrial permeability transition	*SLC25A4*, *HK1*, *VDAC3*, *VDAC1*, *PPID*, *CKMT1A*, *VDAC2*	*PPID* *, CAPN2, VDAC2, SLC25A5*
Mitochondrial genes	*ATP5C1*, *ATP5H*, *ATP5L*, *AIFM1*, *ATP5G1*, *NDUFB6*, *NDUFC2*, *ATP5A1*, *COX7A2*, *NDUFS4*	*ATP5G1*, *NDUFAB1*, *COX5B*, *ATP5B*, *NDUFB9*, *ATP5G3*, *COX6A1*, *COX6C*, *TACO1*, *COX8A*
Pentose phosphate pathway	ACACB, NME4	*NME2* *, ACACB, RRM1, ACSS2, RRM2B, NME4, TXN, TP53, NME7*
Necroptosis	B4GALT6, CUL3, RBX1, RIPK3, HK3, CASP10, PPP1CC	*GUCY1A3* *, H2AFX, PARP1, CFLAR, RIPK3, GUCY1A2, DNM1L, MLKL, NCF1*
Glutamine metabolism	*GLS*, *GLS2*	*MLKL* *, RIPK3, RIPK1, PRPS1*
FAS response	*CFLAR*, *CASP8*, *MLKL*, *GALNT14*, *CASP10*	*PTPN13* *, TNFSF10, MLKL, CFLAR, RIPK1, SQSTM1*
TNF response	*CYLD*, *MAP3K7*, *BIRC2*, *NSMAF*, *TAB2*	*CYLD* *, FLAD1, RNF11, PPP6C, CIAPIN1, RIPK1, CFLAR*
TRAIL response	*CYLD* *, CASP8, MLKL, GALNT14, CASP10*	*MLKL*, *TNFAIP3*, *CFLAR*, *RIPK1*, *TNFRSF10D*
RCD genes	*FBXO45*, *PKM*, *CDC42*, *ATG12*, *HSP90AA1*, *SIRT1*, *HIF1A*, *CASP3*, *BIRC2*	*KLF8* *, BCL2L11, STAT5A, PGK1, HUWE1, ALAS1, ENO1, NPM1, HSP90B1, ELK1 *
Apoptosis	*RHOA* *, TCEB1, CYCS, SKP1, CCNB1, NKRF, BECN1, YWHAG, RPS6KA1, HIP1*	*DIABLO* *, STAT5A, RIPK3, CISH, RPS6KA1, HUWE1, C1QBP, BCL2L11, CDK6, PPP3R1*
DNA damage response	*PPP2R1A*, *CAB39*, *CSNK2A1*, *NKRF*, *HUWE1*, *RNF8*, *PRKDC*	*H2AFX* *, RAD50, MRE11A, ACSS2, HUWE1, BARD1, NKRF, CANK2A1, PPARGC1A, APBB1*
Porphyrin metabolism	*UROD*, *COX10*, *UROS*, *CPOX*, *COX15*	*CPOX* *, ALAS1, HMBS, ALAS2*
ER stress	*DERL1* *, DERL2, HSPA9, ATP2A1, MAP3K5, MAP3K4, DNAJC3, VCP, EIF2S3*	*NLRC4* *, PDIA6, RYR2, NFKB1, PDIA4, DNAJB11, SEC61G, SEC61A1, CREB3L4, ITPR1*
Antioxidant response	*NDUFS1* *, AIFM1, RBX1, CUL3, NDUFC2, NDUFAF1, COX7A2, NDUFS4, NDUFB5, COX6B2*	*UQCRH* *, MCL1, NDUFAB1, CYC1, COX5B, STAT5A, NDUFB9, COX6B1, COX4I2, COX6C*
Oxidative phosphorylation and TCA cycle	*NDUFS1* *, ATP5H, SLC25A14, PARK2, IDH3B, NDUFC2, FH, ATP5G1, IDH3A, SLC1A5*	*ATP5B, NDUFAB1, COX5B, ATP5G3, COX8A, COX6C, COX6B1, FH, NDUFB10, COX5A*

Green: positively contributing genes, red: negatively contributing genes.

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
