# Peer review of "Comprehensive Map of the Regulated Cell Death Signaling Network: A Powerful Analytical Tool for Studying Diseases"

_cancers, 2020, doi:10.3390/cancers12040990_

Round 1
Reviewer 1 Report
The modifications and supplement in this revised manuscript by authors was appreciated, it made this article more significant in scientific research and medical applications. The logical and architecture of this revised manuscript are both fine.
Reviewer 2 Report
I have no further comments to add. The authors have addressed some of the suggestions I have made, or provided appropriate reasons why they are not taking up a particular suggestion
This manuscript is a resubmission of an earlier submission. The following is a list of the peer review reports and author responses from that submission.
Round 1
Reviewer 1 Report
Ravel and Gomey et all built a cell death database which can follow the cell death signal from the initiator layer to the execution layer. They added not just the gens but small molecules and had done extensive and well-documented curation. The interface of the database is based on the lab’s previously built cell map project, where this work fit. I have overall minor comments. The presented expression analysis shows spectacular differences how in different diseases or cancer types the regulated cell death pathways work.
Figure 1 Please when you make the figure make the boxes at the end of the figure with the same margin end and remove the red line under the database names.
Line 605-607 I have not understood the genes are correlating with what. Their expression to the ROMA module score? To the disease state?
Please look up at the literature already existing cell death specific databases and use not just KEGG and Reactome as ana example. The database structure is similar to Rectome’s so the comparison is understandable. For apoptosis I found mostly protein domain interaction and protein list databases (http:\\www.apoptosis-db.org or http://www.deathbase.org/) , but I suggest one more literature search by the authors to check whether there is any additional database in the given cell death processes. For autophagy, the Autophagy Regulatory Database is currently the standard for autophagy research (https://www.ncbi.nlm.nih.gov/pubmed/25635527). Please cite.
Please provide as a Supplementary table the GSEA results and mention them in the example studies. The comparison with existing method is very valuable.
Author Response
Comments and Suggestions for Authors
Ravel and Gomez et all built a cell death database which can follow the cell death signal from the initiator layer to the execution layer. They added not just the gens but small molecules and had done extensive and well-documented curation. The interface of the database is based on the lab’s previously built cell map project, where this work fit. I have overall minor comments. The presented expression analysis shows spectacular differences how in different diseases or cancer types the regulated cell death pathways work.
Response 1: We are glad that the reviewer appreciates the value of the resource and data analysis in the context of the RCD map.
Figure 1 Please when you make the figure make the boxes at the end of the figure with the same margin end and remove the red line under the database names.
Response 2: Thanks for the note, it is corrected.
Line 605-607 I have not understood the genes are correlating with what. Their expression to the ROMA module score? To the disease state?
Response 3: The precision is now added in to the M&M section, paragraph “Module scores calculation and visualization” (lines 653-658):
ROMA quantifies the activity of a group of genes biologically related, referred to as a module in the RCD map (see section” Structure of the Regulated Cell Death map”) , by calculating the largest amount of one-dimensional variance in the expression of the genes that constitute a module across the samples.
The comparison is done in the following mode (added to the M&M section paragraph “Data Acquisition and groups definition”, p. 21):
The Gene expression datasets from lung cancer and Alzheimer’s disease from GEO were grouped the cases of each disease and compared them to their respective controls. The ovarian cancer data was (TCGA), is analysed for disease groups. The group definitions are previously described in PMID: 21720365. We cite this refernce.
Please look up at the literature already existing cell death specific databases and use not just KEGG and Reactome as an example. The database structure is similar to Rectome’s so the comparison is understandable. For apoptosis I found mostly protein domain interaction and protein list databases (http:\\www.apoptosis-db.org or http://www.deathbase.org/) , but I suggest one more literature search by the authors to check whether there is any additional database in the given cell death processes. For autophagy, the Autophagy Regulatory Database is currently the standard for autophagy research (https://www.ncbi.nlm.nih.gov/pubmed/25635527). Please cite.
Response 4: Thanks for drawing our attention to these valuable resources. We now mention them in the text on pp. 10 (lines 317-334). In addition we checked the literature and cite additional related resources. We also explain why the formal comparison of RCD map is done vs. Reactome and KEGG pathways. In fact, these are the only resources where the same paradigm of process representation is applied, that make the comparison meaningful.
Please provide as a Supplementary table the GSEA results and mention them in the example studies. The comparison with existing method is very valuable.
Response 5: The GSEA analysis for the comparison of datasets from lung cancer and Alzheimer’s disease is now added (p.16, lines 462-466; p.22, lines 664-666; Supplementary Table 4 and Supplementary Figure 8). Both methods, ROMA and GSEA show the same enrichment pattern of RCD biological modules.

Reviewer 2 Report
In this report, Ravel et al., provided a map which covers all kinds of cell death signaling pathways from various databases. This map also covered energy, metabolic and mitochondrial mechanisms, besides, the author also applied this map to uncover the different death-related pathways which were occupied between Alzheimer disease and NSCLC. Finally, the authors also applied this map to analyze RCD of subtypes of ovarian cancer database and discovered the distinct death-related pathways among them. Generally, this report mainly focused on how to establish this map using bioinformatic approaches. The authors provide a potential tool to interpret the possible mechanisms of regulated cell death in cancers and disease. The research design, the logical and architecture of this article was well-conducted. The majorly concerned issue regarding this article is the lacking of experimental designs to prove or demonstrate the practicality of this map in serving as a marker maker or therapeutic guide.
Author Response
In this report, Ravel et al., provided a map which covers all kinds of cell death signaling pathways from various databases. This map also covered energy, metabolic and mitochondrial mechanisms, besides, the author also applied this map to uncover the different death-related pathways which were occupied between Alzheimer disease and NSCLC. Finally, the authors also applied this map to analyze RCD of subtypes of ovarian cancer database and discovered the distinct death-related pathways among them. Generally, this report mainly focused on how to establish this map using bioinformatic approaches. The authors provide a potential tool to interpret the possible mechanisms of regulated cell death in cancers and disease. The research design, the logical and architecture of this article was well-conducted.
Response 1: We thank the reviewer for appreciation of our effort.
The majorly concerned issue regarding this article is the lacking of experimental designs to prove or demonstrate the practicality of this map in serving as a marker maker or therapeutic guide.
Response 2: The major role of pathways and networks resources for the therapeutic guidance is providing signatures of processes deregulation. Once the involved pathway are identified, they can be further modelled to predict concrete targets with in these pathways. Of course, the experimental validation in vitro and in vivo need to be performed using e.g. specific inhibitors or knock out techniques.
We now added to the discussion these points ( p. 19, lines 534-541).

Reviewer 3 Report
The manuscript from Ravel et al describes their systems biology approach to build an online tool that can be used to explore regulated cell death pathways (RCD). They then map how RCD map can be used to interpret expression data from certain disease states. The RCD as presented is clearly a considerable exercise and I can certainly appreciate the value of a comprehensive assembly of data from the cell death literature to develop representation of potential or likely pathways of regulation. I do, however, harbor some reservations.
1. Since not all input data to the RCD are equally reliable (or at least equally abundant), one might get the impression that all the modules within the RCD are equally robust and confirmed. Are there measures of reliability or certainty that can be applied to the various modules of the map? Do the authors contend that all features if the map are equally likely to be true? How will readers of the paper and users of the RCD map appreciate levels of certainty about each of the modules?
2. I was not convinced by the data in which public-available expression data from Alzheimer's Disease or non-small cell lung cancer are overlaid on the RCD in order to identify active (or non-active) pathways of RCD in these disease. I could not determine what level of expression difference was required to call a gene differentially expressed, and to what controls each of the gene sets was being compared in order to establish which genes and therefore modules are on or off. It was not clear how many genes needed to be over expressed or under expressed in order to establish the activity of a module in the disease states. Clarification of this section and an explanation of what is being done to infer different RCD in the disease states would be very helpful for readers. At one levels, it seems to undermine the primary motivation for a systems approach to then suggest cell death regulation can be inferred from RNA expression differences alone.
3. From the perspective of potential readers of the paper, Figures 2A, 5B and C, and 6A,B,C and D are challenging. It was difficult to appreciate what these individual figure panels were trying to communicate. It is a challenge to capture the richness and complexity of the RCD map as it exists in html format on the web in a figure, however I urge the authors to consider alternative representations so that others without their expertise can understand the data they are trying to communicate, and will be encouraged to explore the web site and use the resource.
4. I was perplexed by some of the columns in Table 1. I could not, in particular, work out how and why some modules had zero values in the Genes column. I could not find a legend or explanation as to what these values refer to.
5. In the legend of Figure 3, it states "Major players were selected on the RCD map by tags and the scheme was deduced from the map and further manually edited based on the number of interactions on the RCD maps and information from the literature". What is the manual editing? How much manual editing is required to capture bona fide interactions between RCD modules?
Author Response
The manuscript from Ravel et al describes their systems biology approach to build an online tool that can be used to explore regulated cell death pathways (RCD). They then map how RCD map can be used to interpret expression data from certain disease states. The RCD as presented is clearly a considerable exercise and I can certainly appreciate the value of a comprehensive assembly of data from the cell death literature to develop representation of potential or likely pathways of regulation.
Response 0: We thank the reviewer for this positive comment.
I do, however, harbor some reservations.
- Since not all input data to the RCD are equally reliable (or at least equally abundant), one might get the impression that all the modules within the RCD are equally robust and confirmed. Are there measures of reliability or certainty that can be applied to the various modules of the map? Do the authors contend that all features if the map are equally likely to be true? How will readers of the paper and users of the RCD map appreciate levels of certainty about each of the modules?
Response 1.
There are a particular literature selection rules that are followed by the map creator in the majority of the cases. This is explained in the M&M section lines 582-590.
The rules for literature selection are as following: first five to ten reviews articles in the field were chosen for extracting the consensus pathways and regulations accepted in the field and for approaching to the literature suggested in these reviews. This allowed to construct the ‘backbone’ of the network and to define the biological processes boundaries (Figure 1). Further, information from original papers, preferably most recent studies, is analysed and added to enlarge and enrich the network. In majority of cases, the decision about adding a biochemical event or regulation or a process to the map has to be supported by at least two independent studies performed in different teams.
In addition, more formal way is scoring the biochemical reactions on the map by ‘Confidence score”. This approach is described the M&M section lines 603-606 and now also supported by three examples shown in the Supplementary Figure 3. In fact, to evaluate the reliability of the mechanisms represented on the map, each reaction is provided with a confidence score. There are two values indicating the publication score (REF) and the functional proximity score (FUNC) that is computed based on Human Protein Reference Database protein–protein interactions (PPI) network (PMID: 26192618). This confidence score principle has been already published by us (PMID: 26192618).
- I was not convinced by the data in which public-available expression data from Alzheimer's Disease or non-small cell lung cancer are overlaid on the RCD in order to identify active (or non-active) pathways of RCD in these disease. I could not determine what level of expression difference was required to call a gene differentially expressed, and to what controls each of the gene sets was being compared in order to establish which genes and therefore modules are on or off. It was not clear how many genes needed to be over expressed or under expressed in order to establish the activity of a module in the disease states. Clarification of this section and an explanation of what is being done to infer different RCD in the disease states would be very helpful for readers. At one levels, it seems to undermine the primary motivation for a systems approach to then suggest cell death regulation can be inferred from RNA expression differences alone.
Response 2.
We enriched the text with explanations of the method and clarifications. Please see pp. 21-22, lines 651-666. In fact, the enrichment methods ROMA and GSEA were already published, we refer to the corresponding papers where the details of the methods are described.
- From the perspective of potential readers of the paper, Figures 2A, 5B and C, and 6A,B,C and D are challenging. It was difficult to appreciate what these individual figure panels were trying to communicate. It is a challenge to capture the richness and complexity of the RCD map as it exists in html format on the web in a figure, however I urge the authors to consider alternative representations so that others without their expertise can understand the data they are trying to communicate, and will be encouraged to explore the web site and use the resource.
Response 3.
Figure 2A shows the structure of the resource online. The purpose is to make the reader familiar with the webpage layout. It also depicts the top level view of the RCD map with the borders of functional modules, letting the reader to get used to the distribution of processes on the map. The same map level is used for figures 5 and 6.
Figures 5 and 6 aim to demonstrate in a very visual mode the changes in the RCD modules regulation while comparing two disease (Figure 5) or comparing different groups of the same disease (Figure 6). We added the location of the most contributing genes on the modules that are important for the discussion of the results of figure 5.
In addition, Figure 6 shows overlay of two omics data types simultaneously. We clearly explain the glyphs meaning in the figure legend.
Since the resource is a web-platform for data visualization, the user and the readers communicate dynamically, zoom in on the area of interest and explorer the precise reactions and molecules in the area of interest.
- I was perplexed by some of the columns in Table 1. I could not, in particular, work out how and why some modules had zero values in the Genes column. I could not find a legend or explanation as to what these values refer to.
Response 4.
We re-phrased the legend of Table 1, making the point clearer. Zero values indicate absence of the molecule category in the corresponding functional modules of the RCD map.
- In the legend of Figure 3, it states "Major players were selected on the RCD map by tags and the scheme was deduced from the map and further manually edited based on the number of interactions on the RCD maps and information from the literature". What is the manual editing? How much manual editing is required to capture bona fide interactions between RCD modules?
Response 5.
We completely rephrased and extended the Figure 3 legend that now clearly explains the process of network pruning and generation of the diagram shown in this figure.
